# Factors accelerating time to death among persons with Tuberculosis in Western India: Evidence from a community-based retrospective death audit

Harsh Shah[1]*, Jay Patel[1], Somen Saha[1], Bhavesh Modi[2], Pankaj Nimavat[1]

**1** Department of Public Health Science, Indian Institute of Public Health Gandhinagar (IIPHG), Gandhinagar, India, **2** Scientist G & Director, National Institute of Occupational Health, Ahmedabad, India

* hdshah@iiphg.org

## Abstract

### Introduction

Tuberculosis (TB) remains a leading cause of death globally. India aims to eliminate TB by 2025; however, persistently high mortality rates suggest critical failures in early intervention, particularly among vulnerable populations. This study examined the clinical, social, and system factors associated with accelerated mortality among notified persons with TB (PwTB) in Western India.

### Methods

A cross-sectional study was conducted in six districts of Gujarat, India, during 2023–2024 using Community-Based Verbal Autopsy (CBVA) among relatives of 149 deceased PwTB. Sociodemographic, clinical, and TB care cascade data were collected, and a retrospective time-to-event analysis was performed. A Cox proportional hazards model was used to assess the factors associated with a shorter time from diagnosis to death.

### Results

Majority of deceased belonged to 26–50 years age group (40%), with a high male predominance (81.9%). Nearly half (48.3%) had comorbidities, and 65.8% had a history of addiction. A substantial median delay of approximately five weeks was observed between symptom onset and treatment initiation. Following a confirmed diagnosis, the majority of deaths (nearly 80%) occurred within the first 16 weeks. A comparable trend was noted after the start of treatment, with about 78% of fatalities occurring within 15 weeks. In the adjusted Cox regression model, key population status (HR = 1.5, p = 0.01), presence of comorbidities (HR = 2.0, p < 0.001), and

**Data availability statement:** All relevant data are within the paper and its Supporting Information files.

**Funding:** The author(s) received no specific funding for this work.

**Competing interests:** The authors have declared that no competing interests exist.

drug-resistant tuberculosis (HR = 1.7, p = 0.003) were independently associated with a shorter time from diagnosis to death.

## Discussion

The findings highlighted the convergence of clinical complexity, social vulnerability, and persistent system constraints that shorten survival, even after treatment initiation. Strengthening early case diagnosis, integrating comorbidity management, improving health system responsiveness, and implementing targeted strategies for vulnerable populations are critical. A systematic TB Death Surveillance and Response System (TBDSR), integrating facility-, community-based reviews, and digital reporting, would provide actionable insights to inform timely intervention and support progress towards TB elimination in India and other high-burden settings.

## Introduction

Tuberculosis (TB), an ancient and persistent infectious disease, continues to pose a formidable global public health challenge, disproportionately affecting vulnerable populations. Despite significant advancements in prevention and control, TB remains a leading cause of death from a single infectious agent, claiming over a million lives annually and undermining global health security [1]. The global commitment to addressing health disparities is underscored by the Sustainable Development Goals (SDGs), with SDG 3.3 specifically targeting the end of the TB epidemic by 2030. This objective is intrinsically linked to universal health coverage (UHC), essential for equitable access to quality TB care and strengthening resilient health systems [2].

India, a country with the highest global TB burden, accounts for an estimated 26% of the world's TB cases and 34% of global TB deaths [3]. In response, the Government of India has set an ambitious target to eliminate tuberculosis by 2025, five years ahead of the global SDG timeline. This commitment is operationalized through the National Tuberculosis Elimination Program (NTEP), which is aligned with the World Health Organization's End TB Strategy and guided by India's National Strategic Plan (NSP) for Tuberculosis Elimination (2017–2025). The NSP provides the national policy framework for TB control, structured around four strategic pillars—Detect, Treat, Prevent, and Build—emphasizing patient-centered care, strengthened health systems, and intensified research and innovation [4]. Despite these comprehensive efforts, persistent challenges—such as the clinical complexity of TB, comorbidities, diagnostic delays, limited integration of the private sector, the pervasive impact of socio-economic factors and stigma—continue to hinder progress, a situation recently exacerbated by global health crises like the COVID-19 pandemic. [5]. These challenges are not unique to India but resonate across many high-burden, resource-limited settings worldwide [6].

A detailed understanding of the social, clinical, and health system factors associated with tuberculosis mortality is essential for identifying preventable deaths and strengthening programmatic responses. In many resource-limited settings with a

weak vital registration system, tuberculosis mortality surveillance relies heavily on routine clinical records and medical certification of cause of death, which are often incomplete or prone to misclassification [7,8]. As a result, tuberculosis-related deaths—particularly those involving diagnostic delays, deaths occurring outside health facilities, or complex comorbid pathways—may be under reported or inadequately characterized within routine surveillance systems [9,10]. Using a community-based verbal autopsy approach, this study examines factors associated with shortened time to death among persons with tuberculosis (PwTB) in Western India, with the aim of generating evidence to inform policy and strengthen TB control strategies in similar high-burden contexts.

## Methodology

### Study design and settings

A cross-sectional, community-based death audit was conducted among caregivers and relatives of deceased PwTB between October 2023 and December 2024. The study was designed as a cross-sectional audit with the reconstruction of care-cascade timelines utilizing a retrospective cohort approach to evaluate the journey of deceased individuals. Gujarat, a western Indian state with a population of approximately 60.4 million, exhibits substantial epidemiological, geographic, and demographic heterogeneity. The districts for this study were selected using a purposive, two-stage approach. First, priority was given to districts reporting a tuberculosis mortality proportion greater than 5% in the 2022 Ni-kshay surveillance data, with the objective of examining settings where systemic bottlenecks in the TB care cascade were most pronounced and actionable.

Accordingly, six districts—Ahmedabad, Aravalli, Rajkot, Junagadh, Surat, and Dahod—were selected in consultation with the Gujarat State Tuberculosis Department to ensure representation across urban, rural, and tribal contexts. These districts collectively accounted for approximately 65% of TB notifications and 60% of reported TB deaths in the state in 2022. Second, district selection sought to capture variations in healthcare infrastructure, access, and service delivery across the state, which was informed by participatory discussions with government and NTEP experts. Although this purposive sampling strategy emphasizes high-burden districts, the inclusion of geographically and programmatically diverse settings enhances the relevance of findings to a broad range of healthcare contexts within Gujarat.

### Study definition

The study adhered to the established case definitions outlined in the National Tuberculosis Elimination Program (NTEP) guidelines. NI-KSHAY- (Ni = End, Kshay = TB) is a web-enabled patient management system for TB control under the National Tuberculosis Elimination Programme (NTEP). It was developed and is maintained by the Central TB Division (CTD), Ministry of Health and Family Welfare, Government of India [11]. The classification of deceased individuals included three primary categories: new cases (those on a drug-sensitive regimen), retreatment cases (drug-sensitive patients with a history of a previous TB episode), and Programmatic Management of Drug-resistant Tuberculosis (PMDT) cases (individuals on a drug-resistant treatment regimen). The Key Population encompasses patients identified by the NTEP guidelines as having an increased vulnerability or risk for tuberculosis due to various factors. This group includes individuals such as contacts of persons with TB, diabetics, tobacco users, prison inmates, miners, migrants, refugees, urban slum dwellers, and healthcare workers.

### Study population and sampling methods

The study population for the community-based death audit comprised family members or caregivers of deceased PwTB who were knowledgeable about the patient's care-seeking and treatment journey from the onset of symptoms to death. Eligible respondents were identified based on their relationship with the deceased and their direct involvement in the care process, including accompanying the patient for a minimum of three healthcare visits and having adequate awareness

of the clinical course. Respondent eligibility was assessed by trained data collectors to ensure the reliability of reported information.

According to Ni-kshay surveillance data for 2022, Gujarat reported approximately 151,912 tuberculosis cases across the public and private sectors, with 8,380 deaths, corresponding to a mortality proportion of 5.5%. The sample size was calculated using the standard formula for estimating a population proportion, assuming a 5% mortality proportion, 95% confidence level, a 5% margin of error, design effect of 1.5 to account for district-level clustering, and 10% non-response rate. Using the formula $N = Z^2_{(1-\alpha)/2} P(1-P)/\varepsilon^2$, this yielded a final sample size of 149 cases. Quarterly tuberculosis notification registers for October 2023–December 2024 from the selected districts were extracted from the Ni-kshay portal, and all notified deaths were line-listed. Simple random sampling was chosen to ensure each sample had an equal probability of selection, thereby reducing selection bias within high-burden districts. This method ensured the findings were representative of the mortality patterns within the selected districts.

The final eligible tuberculosis cases were identified based on predefined inclusion and exclusion criteria. The inclusion criteria were as follows: (i) notified in the Ni-kshay system whose place of treatment was within the selected districts of Gujarat; (ii) PwTB whose treatment outcome was recorded as "Died" after treatment initiation; and (iii) availability of a family member or close caregiver who provided informed consent to participate in the community-based verbal autopsy interview. The exclusion criteria included cases in which families had migrated outside the study area, were untraceable, or did not reside within the selected public health institution (PHI) catchment areas at the time of the survey, as well as cases in which eligible respondents declined to provide consent.

## Data collection procedures and tool validation

Quantitative data were collected using a predefined and pretested Community-Based Verbal Autopsy (CBVA) instrument. The standard World Health Organization verbal autopsy framework was adapted in collaboration with the Gujarat State National Tuberculosis Elimination Program (NTEP) team to enable a detailed assessment of the tuberculosis care cascade. Key adaptations included the incorporation of care-cascade timestamps capturing the period from symptom onset to diagnosis and treatment initiation, as well as items documenting healthcare-seeking pathways across public, private, and traditional care providers. Additional modules were included to record NTEP-specific treatment adherence monitoring mechanisms and selected social risk factors, such as addiction history and key population status.

To ensure data quality and consistency, interviews were conducted by trained medical officers with public health backgrounds who received structured training on interview procedures, ethical considerations, and standardized use of the CBVA instrument. Potential recall bias was minimized by selecting primary caregivers as the respondents. Where available, proxy-reported dates—including symptom onset, diagnosis, and treatment initiation—were cross-verified against treatment cards and digital Ni-kshay records to enhance data accuracy.

The modified CBVA captured information across three domains: (i) socio-clinical characteristics, including demographic, socio-economic, behavioral factors, and comorbidities; (ii) TB care cascade indicators, including healthcare utilization patterns, diagnostic and treatment delays, and treatment adherence options; and (iii) circumstances preceding death, including terminal symptoms, warning signs, and the place and mode of death. Key dates related to diagnosis, treatment initiation, and death were extracted to support retrospective time-to-event analysis. All interviews were conducted with cultural sensitivity and respect for bereaved families.

## Data analysis and statistical modeling

The data were systematically managed and analyzed using IBM SPSS Statistics for Windows, Version 25.0. Descriptive statistics, including means, medians, ranges, frequencies, and percentages, were employed to summarize the demographic, clinical, and care-cascade characteristics of the deceased PwTB. Variables such as Body Mass Index (BMI) and clinical severity at the time of diagnosis were not included in the final analysis as they were not consistently recorded in

the secondary treatment records or reliably recalled during verbal autopsies. To ensure technical appropriateness in the retrospective analysis, time intervals across the care cascade—specifically from symptom onset, first consultation, diagnosis, and treatment initiation to the date of death—were calculated using raw date-specific timestamps to derive median and interquartile (IQR) range durations.

A Cox Proportional Hazards Model was utilized to analyze the time from diagnosis to death for the cohort of 149 deceased PwTB. Since the study was a death audit where all participants had reached the final event, no censoring was required for the analysis. The model assessed the influence of covariates—including key population status, presence of comorbidities, addiction history, type of TB (PMDT vs. non-PMDT), disease site, and age—on the hazard of accelerated mortality. The proportional hazards assumption was verified using the Schoenfeld global test (p = 0.21). Results are reported as Hazard Ratios (HR) with 95% Confidence Intervals (CI) and associated Wald p-values. Variables such as BMI and severity at diagnosis were not captured in verbal autopsies; results from the Cox model should be interpreted within this clinical constraint.

### Ethical considerations

The study adhered to ethical standards, with all participants fully informed about the study's purpose, procedures, potential risks and benefits, and their rights. Participation was voluntary, and written informed consent was obtained after addressing any questions. Confidentiality and anonymity were maintained through secure data handling and the de-identification of personal information. This study was approved by the Institutional Ethics Committee of the Indian Institute of Public Health, Gandhinagar. (IEC – TRC-IEC No: 05/2023–24)

## Results

This study analysed 149 TB-related deaths, revealing distinct patterns in socio-demographic characteristics, comorbidities, healthcare-seeking behaviours, disease progression timelines, and treatment adherence.

### Sociodemographic characteristics

Most tuberculosis-related deaths occurred among individuals aged 26–50 years (40.3%), followed by those aged 51–65 years (32.9%). Persons aged over 65 years accounted for 18.1% of deaths, while younger age groups—19–25 years and 0–18 years—contributed 6.7% and 2.0%, respectively. The median age at death was 50 years (interquartile range [IQR]: 35–65), and the mean age was 50.7 years (standard deviation: 19.5), with ages ranging from 17 to 95 years. Of the deceased PwTB, the gender distribution was observed that 81.9% (n = 122) were male and 18.1% (n = 27) were female.

### Clinical characteristics

Pulmonary tuberculosis was the predominant disease form, reported in 88.6% (n = 132) of cases. Most deceased individuals were classified as new, drug-sensitive TB cases (64.4%, n = 96), followed by re-treatment cases (26.8%, n = 40), while 8.7% (n = 13) were managed under the Programmatic Management of Drug-resistant Tuberculosis (PMDT). Treatment adherence was monitored through National Tuberculosis Elimination Program (NTEP) mechanisms, including Directly Observed Treatment (DOT) for 59.7% (n = 89) of cases and digital adherence technologies (99DOTS or MERM) for 40.3% (n = 60). Despite the presence of these monitoring systems, 36.9% (n = 55) of persons with tuberculosis (PwTB) were reported as non-adherent prior to death.

Overall, 63.1% (n = 94) of deceased PwTB belonged to key populations identified as vulnerable under NTEP guidelines. Comorbid conditions were documented in 48.3% (n = 72) of deaths, with diabetes being the most reported comorbidity (10.8%, n = 16), frequently co-occurring with hypertension (7.3%, n = 11) and liver disease (6.7%, n = 10). Chronic obstructive pulmonary disease was reported in 6.7% (n = 10) of cases and HIV co-infection in 2.0% (n = 3). Behavioral risk factors were common, with 65.8% (n = 98) of deceased PwTB reporting a history of substance use. Among these, tobacco use

alone was documented in 38.8% (n = 38), alcohol use alone in 16.3% (n = 16). Notably, the largest proportion of this cohort was characterized by polysubstance (tobacco and alcohol) use (n = 44), indicating that nearly 45% of those with addictions were engaged in the concurrent use of both substances.

## First point of contact and frequency of consultations for TB diagnosis

Analysis of the first point of contact for symptoms among the PwTB showed varied care-seeking behavior. Among the 149 PwTB, 47.8% (n = 71) initially sought care at public health institutions, 31.6% (n = 47) at private healthcare providers, and 10% (n = 15) at AYUSH (traditional medicine) facilities. A small proportion consulted local healers (1.3%, n = 2), while 9.3% (n = 14) were unable to specify their first point of contact.

The complexity of the diagnostic care pathway was assessed using the number of healthcare facilities visited prior to tuberculosis confirmation. Among the 149 PwTB, 40.3% (n = 60) received a diagnosis at their first point of contact. In contrast, 20.1% (n = 30) visited two facilities, 11.4% (n = 17) visited three facilities, and 15.4% (n = 23) visited more than three facilities before diagnosis. An additional 12.8% (n = 19) were unable to recall the number of facilities visited prior to diagnosis.

## Place of deaths

A substantial number of deaths (69.1%, n = 103) occurred at home. Only 21.5% (n = 32) died within a health facility, and 9.4% (n = 14) died in transit while seeking care. During the final health event, death, 69.1% (n = 103) of individuals did not utilize any transport services. Among those who attempted to reach a healthcare facility, 18.8% (n = 28) used an ambulance and 12.1% (n = 18) were transported via private or personal vehicles.

## Delay in diagnosis and pre-treatment care pathway

Substantial delays were observed across the pre-diagnostic and pre-treatment phases of the tuberculosis care cascade (Table 1). During the analysis, it was observed that median time from symptom recognition to first formal healthcare consultation and to confirmed tuberculosis diagnosis was approximately five weeks reflecting additional delays within the diagnostic process following first contact with the health system which is longer compared to Indian standard guidelines to diagnose a TB case within two weeks of common TB symptoms appearance. The median time from symptom onset to treatment initiation was 34 days (IQR: 23–55 days), indicating most patients started anti-TB treatment after one month. Although treatment began shortly after diagnosis, the delay from symptom onset to treatment remained substantial, with wide interquartile ranges indicating considerable variability. These findings showed that prolonged pre-diagnostic and pre-treatment intervals were common among deceased PwTB.

## Disease progression and time to death after diagnosis

The study highlighted a stepwise compression of survival across the care cascade, highlighting rapid mortality following diagnosis and treatment initiation among deceased persons with tuberculosis. Fig 1 shows the distribution of time to death

**Table 1. Timeline to death (event) – onset of symptoms to death – days and weeks.**

| Care Cascade Timestamps | Days (Median, IQR) | Weeks (Median, IQR) |
| --- | --- | --- |
| Onset of Symptoms to First Consultation | 21 (12–32) | 3.0 (1.7–4.6) |
| Onset of Symptoms to Confirmed Diagnosis | 31 (22–52) | 4.4 (3.1–7.4) |
| Onset of Symptoms to Treatment Initiation | 34 (23–55) | 4.9 (3.3–7.9) |
| Onset of Symptoms to Death | 94 (58–146) | 13.4 (8.3–20.9) |

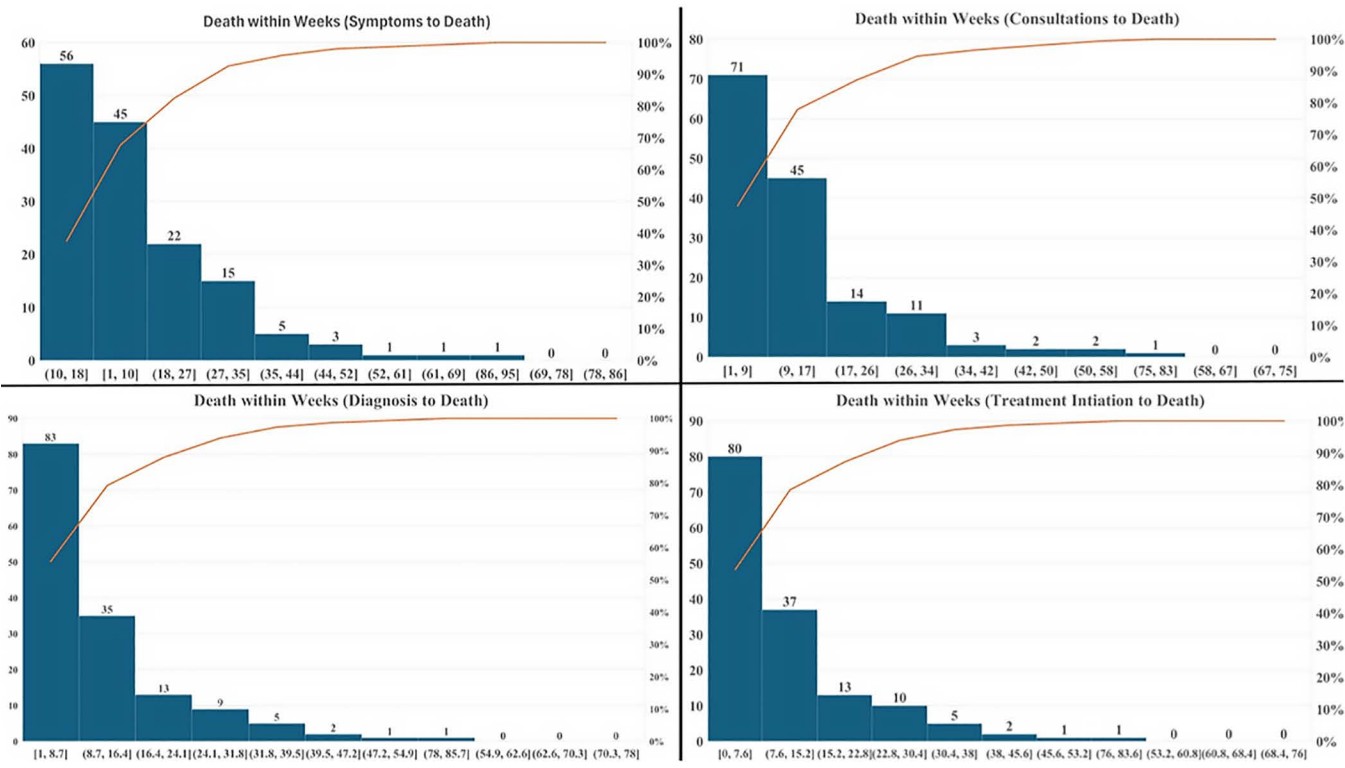

**Fig 1. Time-to-event analysis of tuberculosis mortality (in Weeks).**

across successive stages of the tuberculosis care cascade. From symptom onset to death, 67% of deaths occurred within 18 weeks, and following first healthcare consultation, mortality was further compressed, with approximately 78% of deaths occurred within 17 weeks. After a confirmed diagnosis, deaths clustered even earlier as nearly half of deaths occurring within nine weeks. A similar pattern was observed following treatment initiation, where approximately 78% of deaths occurred within 15 weeks, and more than half within 8 weeks.

### Time to death stratified by sociodemographic and clinical characteristics

Table 2 presents descriptive summaries of the median and range of time intervals from key care cascade events to death, stratified by selected sociodemographic and clinical characteristics. Substantial heterogeneity was observed across strata in the timing of death following symptom onset, first consultation, diagnosis, and treatment initiation.

Shorter median durations from treatment initiation to death were observed among persons with drug-resistant tuberculosis, those with documented comorbidities, and individuals classified as key populations. For example, the median survival following treatment initiation was six weeks among persons with drug-resistant tuberculosis and five weeks among those with comorbidities, compared with nine to ten weeks among their respective reference groups. Similar patterns were observed for the interval between diagnosis and death across these subgroups.

Across most categories, the interval from symptom onset to death remained substantially longer than the post-diagnosis and post-treatment intervals, indicating that a large proportion of total survival time accrued prior to diagnosis. These stratified descriptive findings provided context for the observed clustering of deaths in the early post-diagnosis period and supported the adjusted associations identified using the Cox proportional hazards model.

**Table 2. Median and range of time intervals by selected variables (N = 149) (Weeks).**

| Variable | Category (n) | Symptoms to Death | Consultation to Death | Diagnosis to Death | Treatment Initiation to Death |
|---|---|---|---|---|---|
| Age group (years) | 0–18 (3) | 7 (5–42) | 2 (1–14) | 1 (1–8) | 1 (1–7) |
| | 19–25 (10) | 15 (4–30) | 11 (1–27) | 9 (1–26) | 9 (1–26) |
| | 26–50 (60) | 10 (1–52) | 7 (1–49) | 6 (0–49) | 5 (0–49) |
| | 51–65 (49) | 12 (1–64) | 8 (1–54) | 7 (0–40) | 6 (0–40) |
| | > 65 (27) | 13 (1–87) | 11 (1–83) | 9 (0–81) | 9 (0–81) |
| TB case type | New (96) | 14 (1–88) | 11 (1–83) | 10 (1–82) | 10 (0–81) |
| | Retreatment (40) | 13 (2–52) | 8 (1–52) | 8 (1–38) | 8 (1–35) |
| | PMDT (13) | 12 (3–53) | 6 (1–42) | 6 (1–31) | 6 (1–30) |
| Key population | Yes (94) | 13 (1–65) | 10 (1–60) | 6 (1–45) | 6 (1–40) |
| | No (55) | 15 (1–90) | 12 (1–85) | 9 (1–70) | 9 (1–65) |
| Comorbidities | Yes (72) | 12 (1–60) | 8 (1–55) | 5 (1–40) | 5 (1–35) |
| | No (77) | 16 (1–95) | 13 (1–90) | 10 (1–75) | 10 (1–70) |
| Addiction | Yes (98) | 13 (1–70) | 10 (1–65) | 7 (1–50) | 7 (1–45) |
| | No (51) | 15 (1–85) | 12 (1–80) | 9 (1–65) | 9 (1–60) |
| Type of TB | PMDT (13) | 12 (3–53) | 6 (1–42) | 6 (1–31) | 6 (1–30) |
| | Non-PMDT (136) | 14 (1–88) | 10 (1–83) | 9 (1–82) | 9 (1–81) |
| Site of disease | Pulmonary (132) | 14 (1–88) | 11 (1–83) | 7 (1–82) | 7 (1–81) |
| | Extrapulmonary (17) | 13 (1–70) | 10 (1–65) | 8 (1–60) | 8 (1–55) |

## Results of a cox proportional hazards model

A Cox proportional hazards model was used to examine the association between selected covariates and time from tuberculosis diagnosis to death (Table 3). There was no evidence of violation of the proportional hazards' assumption (Schoenfeld global test p = 0.21), and multicollinearity among covariates was low (variance inflation factors ranging from 1.04 to 1.15). As all 149 deceased PwTB experienced the event of interest (death), the events-per-variable ratio was 24.8.

In the adjusted analysis, the presence of comorbidities was associated with a two-fold higher hazard of death (HR = 2.00; 95% CI: 1.43–2.80; p < 0.001). Drug-resistant tuberculosis (PMDT) was associated with a 70% higher hazard of death compared with non-PMDT disease (HR = 1.70; 95% CI: 1.18–2.45; p = 0.004). Individuals belonging to key populations had a 50% higher hazard of death (HR = 1.50; 95% CI: 1.11–2.03; p = 0.008). Increasing age was also associated with higher mortality, with each additional year corresponding to a 2% increase in hazard (HR = 1.02; 95% CI: 1.01–1.03; p < 0.001). A history of addiction (HR = 1.30; 95% CI: 0.92–1.84; p = 0.138) and site of disease (pulmonary vs. extrapulmonary; HR = 1.10; 95% CI: 0.73–1.66; p = 0.652) were not significantly associated with time to death in the adjusted model.

**Table 3. Results of cox proportional hazards model.**

| Covariate | DF | β (SE) | HR | 95% CI for HR | Wald p-value | Lik. Ratio p-value |
|---|---|---|---|---|---|---|
| Age (per year) | 1 | 0.020 (0.005) | 1.02 | 1.01–1.03 | <0.001 | <0.001 |
| Key Population (Yes vs. No) | 1 | 0.405 (0.153) | 1.5 | 1.11–2.03 | 0.008 | 0.008 |
| Comorbidities (Yes vs. No) | 1 | 0.693 (0.150) | 2 | 1.43–2.80 | <0.001 | <0.001 |
| Addiction (Yes vs. No) | 1 | 0.262 (0.177) | 1.3 | 0.92–1.84 | 0.138 | 0.138 |
| PMDT vs. non-PMDT | 1 | 0.531 (0.187) | 1.7 | 1.18–2.45 | 0.004 | 0.004 |
| Pulmonary vs. Extrapulmonary | 1 | 0.095 (0.211) | 1.1 | 0.73–1.66 | 0.652 | 0.652 |

The selection of variables for stratified analyses and multivariable adjustment was guided by a priori clinical relevance and existing evidence on tuberculosis mortality. Other potentially relevant factors, including nutritional status, unmeasured clinical severity indicators and contextual determinants at the time of diagnosis and treatment initiation, could not be included due to data limitations inherent to verbal autopsy–based designs and are acknowledged as residual confounders in the study limitations.

## Discussion

This community-based study provided a retrospective time-to-event analysis of factors influencing early mortality, among notified PwTB in Western India, a region critical to India's elimination goals. The findings suggested that deceased PwTB were characterized by limited survival following entry into care, consistent with patterns reported in previous studies from various regions of India and other high-burden settings [12–16]. The findings highlighted the co-occurrence of clinical complexity, social vulnerability, and persistent health system constraints that dramatically shortened survival, even after treatment initiation.

### Deconstructing the diagnostic abyss: Failures in the pre-treatment cascade

Despite the programmatic emphasis on rapid treatment, this study identified substantial delays across the pre-diagnostic phase of the tuberculosis care cascade. These delays reflect both patient and health system–level constraints and are consistent with patterns reported from other resource-limited and high-burden settings [17,18].

While programmatic mechanisms were generally effective in linking diagnosed cases to treatment, the diagnostic process itself remained vulnerable to fragmentation. This process was severely hindered by care fragmentation, as nearly half of the deceased PwTB visited two or more facilities before receiving a confirmed diagnosis of TB. The diverse initial points of contact and multiple visits to healthcare providers, underscored the difficulty in creating a seamless diagnostic pathway, thereby impeding early case detection which was highlighted in several studies related to delay in care cascade: particularly in low resource settings [19–21]. These findings reinforce the importance of strengthening diagnostic pathways to reduce delays and improve continuity of care.

### The lethality of late intervention: Factors accelerating mortality

A key finding of this analysis was the identification of a rapid post-diagnosis mortality pattern among specific high-risk subgroups. The Cox proportional hazards model indicated that the presence of comorbidities was associated with a two-fold increase in the hazard of death, while enrolment under the Programmatic Management of Drug-resistant Tuberculosis (PMDT) was associated with a substantially higher hazard than drug-sensitive cases. In addition, individuals belonging to key populations experienced significantly elevated mortality risk. These observations aligned with existing evidences demonstrating the compounded influence of multi-morbidity, treatment complexity, and socioeconomic vulnerability on tuberculosis mortality [22,23].

### Policy imperatives: Strategic reorientation for TB elimination

These findings compel a strategic shift in the NTEP toward an accelerated risk-based intervention model to successfully meet the 2025 elimination goal.

- Policies must mandate immediate clinical risk stratification at the time of diagnosis. This protocol should trigger enhanced integrated care for PMDT patients and those with comorbidities, recognizing that managing these co-factors is essential for survival. This strategy aligns with WHO recommendations emphasizing integrated care and strengthened clinical pathways for comorbid PwTB [24].

- To compress the substantial pre-diagnosis lag, resources must be dedicated to strengthening the availability of rapid molecular diagnostics closer to the community. This is vital for reducing diagnostic facility hopping and accelerating diagnosis, which has proven critical in other high-burden contexts [25–27].

- The high proportion of deaths occurring at home (69.1%) and the limited use of emergency transport (18.8% used ambulances) signify a critical gap in accessible emergency and palliative care services. The NTEP should integrate palliative care services and robust emergency referral protocols for high-risk patients who fail to respond to treatment.

- The overwhelming male predominance in mortality, which was associated with high addiction rates, highlighted the need for targeted male health engagement strategies, mirroring recommendations from mortality audits conducted in other high-burden countries [28,29].

- A crucial overarching recommendation stemming from this research is the imperative to establish a systematic TB Death Surveillance and Response System (TBDSR). This system should integrate Community-Based Death Reviews (CBDRs) and Facility-Based Medical Audits (FBMAs) with digital reporting through platforms such as the Ni-kshay portal, complete with timestamps. The insights provided in this study are primarily intended to inform intervention strategies in high-risk environments where mortality surveillance and response systems require the most urgent attention. The structured audits would provide actionable, real-time insights into specific bottlenecks in the TB care cascade [13].

### Study strengths and limitations

A primary strength of this research was the implementation of a modified Community-Based Verbal Autopsy (CBVA), which successfully captured mortality events occurring outside formal health facilities and facilitated the construction of a detailed care-cascade chronology often missed by routine surveillance. Furthermore, the application of a retrospective time-to-event analysis provided a robust measure of the clinical and systemic factors influencing the hazard of death. Despite these strengths, several limitations must be acknowledged.

The purposive selection of districts with mortality rates exceeding 5% may have introduced selection bias, potentially overestimated the prevalence of specific mortality determinants and limited generalizability to low-burden regions. Additionally, because the study focused exclusively on notified cases, the findings likely underestimated the true mortality burden by excluding undiagnosed or non-notified TB deaths. The methodology was also subject to recall bias regarding precise dates of symptom onset and consultation chronology, which relied on the memory of proxy respondents.

Another limitation of this study which could have contributed to treatment outcome was the absence of validated data regarding concurrent treatment for comorbidities, as available records were fragmented and lacked sufficient detail for cross-verification. Finally, the cross-sectional nature of the data collection restricts definitive causal inference, and while the Cox Proportional Hazards Model adjusted for key covariates, the analysis may be influenced by unmeasured confounders, such as nutritional status (BMI) and granular socioeconomic nuances, which were not available in the audited records [30].

### Conclusion

This study provides compelling evidence that early TB mortality in high-burden settings is not simply a function of diagnostic failure, but a complex outcome driven by the accelerated hazard of death among vulnerable patients with comorbidities and drug resistance. A key recommendation is the establishment of a TB Death Surveillance and Response System (TBDSR) that integrates Community-Based Death Reviews and Facility-Based Medical Audits with digital reporting through the Ni-kshay portal. These audits would provide actionable insights into TB care cascade bottlenecks. This system, with digital reporting and analysis, would enable policymakers to tailor interventions and allocate resources efficiently to reduce TB mortality across India, thereby advancing progress toward TB elimination. Achieving TB elimination by 2025 demands a proactive shift from merely treating TB to aggressively managing the entire syndemic complexity that drives mortality rates.

## Supporting information

**S1 File. Additional file 1 Supplementary Data TB Death Audit.**
(DOCX)

## Acknowledgments

We sincerely thank all the participants who generously shared their experiences during the study. We gratefully acknowledge the continuous support of the District and State TB Cells, Government of Gujarat. Special thanks to the expert team at the Indian Institute of Public Health, Gandhinagar for their valuable guidance and feedback on the final manuscript.

## Author contributions

**Conceptualization:** Harsh Shah, Somen Saha.

**Data curation:** Harsh Shah, Jay Patel, Pankaj Nimavat.

**Formal analysis:** Harsh Shah, Jay Patel.

**Investigation:** Harsh Shah.

**Methodology:** Harsh Shah, Jay Patel, Somen Saha, Bhavesh Modi.

**Project administration:** Harsh Shah.

**Resources:** Harsh Shah, Jay Patel, Somen Saha, Pankaj Nimavat.

**Software:** Harsh Shah.

**Supervision:** Harsh Shah, Somen Saha, Bhavesh Modi, Pankaj Nimavat.

**Validation:** Harsh Shah, Jay Patel, Somen Saha, Bhavesh Modi, Pankaj Nimavat.

**Visualization:** Harsh Shah, Jay Patel.

**Writing – original draft:** Harsh Shah, Jay Patel.

**Writing – review & editing:** Somen Saha, Bhavesh Modi, Pankaj Nimavat.

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
