## [Decision Letter · Decision Letter 0]

10 Dec 2025

Dear Dr. Shah,

**major methodological, structural, and reporting issues**

We look forward to receiving your revised manuscript.

Kind regards,

Yatin N. Dholakia, MD

Academic Editor

PLOS One

Journal Requirements:

Additional Editor Comments:

Authors have reported on the factors accelerating time to death among notified TB cases through a community based death audit in Western India. This is important and policy-relevant, assessing the relevance of TB early mortality surveillance with use of verbal autopsy.

However, the reviewers have identified several major methodological, structural, and reporting issues that must be addressed before the manuscript can be considered further. The required revisions largely involve clarifying the study design, correcting timeline inconsistencies, strengthening methodological reporting, addressing potential biases, improving model diagnostics, and enhancing clarity and consistency.

Reviewers' comments:

Reviewer's Responses to Questions

**Comments to the Author**

1. Is the manuscript technically sound, and do the data support the conclusions?

Reviewer #1: Yes

Reviewer #2: Yes

2. Has the statistical analysis been performed appropriately and rigorously?

Reviewer #1: No

Reviewer #2: Yes

3. Have the authors made all data underlying the findings in their manuscript fully available?

Reviewer #1: Yes

Reviewer #2: No

4. Is the manuscript presented in an intelligible fashion and written in standard English?

Reviewer #1: Yes

Reviewer #2: Yes

Reviewer #1: Overall, the manuscript addressed an important public health issue of early mortality among tuberculosis patients (PwTB) in a high-burden Indian state. The use of community-based verbal autopsy (CBVA) linked with timeline and Cox regression analysis offers meaningful insights for TB mortality surveillance. The study is relevant to the Indian context, aligned well with the policy and can be good evidence for Tuberculosis related literature. However, some methodological and structural issues need be addressed to strengthen the manuscript for publication.

1. The study is described as cross-sectional, but the analytical approach like time-to-event, Cox model and timelines from longitudinal data need to be aligned with traditional cross-sectional definitions. The study seems to be a "retrospective cohort using verbal autopsy data". Please justify its classification as cross-sectional study.

2. The sampling frame is unclear as it says that data was taken from "Quarterly notification registers 2023–24," but verbal autopsies include deaths from 2022–2023. Inclusion period appears inconsistent. Please specify clear timelines of notification period, death period, Interview-verbal autopsy period.

3. Based on >5% death reporting, the study districts were purposively selected. This may overestimate mortality-related determinants. Comparison is needed with the baseline population of all TB deaths or general PwTB. A brief justification is needed about why these 6 districts were selected and do they adequately represent Gujarat. Comment about potential selection bias and its effect on generalizability.

4. Please detail about the Reliability and Validity of Verbal autopsy tool as modifications were done in WHO tool. Please specify what items were adapted, whether tool was validated locally, inter-rater reliability assessments of data collectors.

5. Please discuss about potential confounders and how they were adjusted in the Cox model. acknowledge the unmeasured confounders.

6. Discuss about how the recall bias was mitigated.

7. Results: Table 3 lacks key diagnostics such as Number of events per variable, Model fit indices, Multicollinearity tests.

8. Discussion: While associations are strong, the discussion often implies causality. Modify the language suitably. Avoid implying predictive models unless validated.

9. TBDSR recommendations are relevant but require links to recent state/national policy efforts. Shorten and cite examples from West Bengal, South Africa, or WHO TB mortality audits.

10. Several sentences are long or repetitive. Please maintain consistent terms: PwTB / TB patients / deceased individuals.

11. Ethical Statements: Two contradictory ethical statements can be seen, one in manuscript body and one in declarations section with different approval numbers and dates. Please recheck and correct.

12. Reference formatting is inconsistent. Please modify as per journal guidelines.

Reviewer #2: This manuscript addresses an important public health issue and provides valuable insights. Overall, the paper is a good attempt but some major revisions are required.

Introduction:

Line 78 mentions “qualitative insights from diverse stakeholders”. But this is not included in the paper. Please consider removing this.

Line 81: In the objective, please specify TB - “to explore the sociodemographic, clinical, and health system related factors associated with early mortality in TB patients (or patients notified with TB) in Western India through….."

Methodology:

Line 22, 85 to 87: Time-frame discrepancy.

Abstract and methodology mentions time frame as 2022 to 2023 but from patients notified in NIKSHAY from October to December 2024. Please clarify.

Line 91 What is the status of TB notification in the state and the selected districts?

Line 111 “The study population for the community-based verbal autopsy (CBVA) form (quantitative) included…..” Is there a qualitative component also?

Line 136 What were the modifications in the CBVA tool? How was data accuracy confirmed?

How was cause of death verified? Can there be a misclassification bias?

Line 119: “qualitative variable”…please remove this or clarify.

Line 124. How many cases were line listed? Why was simple random sampling chosen? Does this affect the selection from the 6 districts? Please justify

Line 157 Can you justify using cox proportional methods in this cross-sectional study?

Results:

Line 178, 179 Is there a reason for mentioning median and mean?

Age groups classification strategy?

187, 188 Despite monitoring mechanisms such as Directly Observed Treatment (DOT) (59.7%) and 99DOTS (39.6%), 36.9% (n=55) were reported as non-adherent. Please explain 59.7% and 39.6%? Were all patients notified in NIKSHAY not covered under DOTS?

190 Can the authors provide some more data regarding the key population characteristics?

193 Please specify how many had hypertension or liver disease.

195 Is there any separate data on how many patients were using tobacco? Can the authors justify combining tobacco and alcohol use instead of highlighting smoking/tobacco use alone?

203 , “40.3% (n=60) received a diagnosis at their first point of contact.” Is there data available on how many of these were from public health facilities? Was there a difference in facility hopping in government vs others? This can be used to justify strengthening of public health system.

218 Why were median time from symptm onset to first formal consultation, median delay from symptom onset to diagnosis etc calculated as difference between other variables? Can this be justified? Was raw data not available? Please confirm if it is statistically correct (technical appropriateness) to calculate this way.

223. What is the distribution of the data? Can only median be used? Most of the discussion is pertaining to median.

231 “some PwTB experiencing delays exceeding 5 months”….where is the data for this?

280 “pulmonary site”- please consider changing this. (Pulmonary TB/ site being pulmonary)

Discussion

324 Is there any data on how many patients were on treatment for the comorbidities?

Limitations

357 Please mention that since this study is only on notified TB cases, missed TB deaths (undiagnosed and not-notified) may be higher.

**Do you want your identity to be public for this peer review?** For information about this choice, including consent withdrawal, please see our Privacy Policy

Reviewer #1: **Yes:** Dr. S.Z. Quazi

Reviewer #2: No

---

## [Author Response · Author response to Decision Letter 1]

3 Jan 2026

Dear Academic Editor and Reviewers,

We sincerely thank the Academic Editor and the reviewers (1&2) for their careful assessment of our manuscript and for the constructive and insightful comments provided with minute details. We greatly appreciate the time and effort invested in reviewing our work, which has substantially strengthened the quality, clarity, and rigor of the manuscript.

We have carefully considered each comment and have revised the manuscript accordingly. In response to the reviewers’ suggestions, we have undertaken a comprehensive revision that includes refinement of the sections: study title, introduction, clarification of methodological details, strengthening of the results presentation, and improved alignment between the Results and Discussion sections. Particular attention was given to improving the clarity of the time-to-event analyses, the interpretation of care-cascade delays and early mortality, and the presentation of figures and tables to ensure consistency and transparency.

We have also revised the abstract and discussion to better reflect the study findings without over-interpretation, ensured consistency in terminology and timelines across all sections, and addressed comments related to data availability, ethical considerations, and reporting standards. The data availability statement has been updated to clearly explain ethical restrictions on public data sharing and to outline a transparent mechanism for controlled data access upon reasonable request. Additionally, the reference list has been carefully revised to comply fully with the journal’s Vancouver referencing style.

All changes made in response to the Academic Editor’s and reviewers’ comments are reflected in the revised manuscript. We believe that these revisions have significantly improved the manuscript and have addressed all concerns raised during the review process.

We are grateful for the constructive guidance provided and hope that the revised version meets the expectations of the journal. We would be pleased to provide any further clarifications or revisions if required.

Thank you once again for your thoughtful review and consideration.

Sincerely,

Harsh Shah

(On behalf of all authors)

We sincerely thank you for your detailed guidance and for the opportunity to revise our manuscript. We have carefully addressed each of the points raised and outline our responses below.

1. Compliance with PLOS ONE style and formatting requirements

We have revised the manuscript to fully comply with PLOS ONE style guidelines, including formatting of the main text, title page, author affiliations, tables, figures, and file naming conventions, in accordance with the official PLOS ONE templates provided. We have ensured consistency in font type, font size, line spacing, heading structure, reference style (Vancouver), and figure/table placement as per journal requirements.

2. Data sharing and Data Availability statement

We acknowledge the importance of data sharing and transparency. This study involved community-based verbal autopsies and includes sensitive human participant data such as detailed clinical timelines, health-seeking behaviours, and household-level socioeconomic information. Although all direct personal identifiers were removed prior to analysis, the combination of variables and small geographic units could permit indirect identification of participants.

Accordingly, ethical restrictions on public data sharing were imposed by the Institutional Ethics Committee of the Indian Institute of Public Health Gandhinagar, which approved the study. In line with PLOS ONE policy, we have updated the Data Availability statement to clarify that de-identified data can be made available upon reasonable request, subject to ethics approval and data protection requirements.

Requests for data access may be directed to corresponding author.

As such, option (b) does not apply, and the dataset has not been uploaded to a public repository due to these ethical constraints. The Data Availability statement in the submission system has been updated accordingly.

3. Supporting Information captions and citations

We have added captions for all Supporting Information files at the end of the manuscript, following PLOS ONE Supporting Information guidelines. All in-text citations referring to Supporting Information have been checked and updated to ensure accurate cross-referencing.

4. Reviewer-suggested citations

We carefully reviewed all publications suggested by the reviewers. Relevant references that strengthened the contextualization and interpretation of our findings have been incorporated into the revised manuscript. Citations that were not directly applicable to the study objectives or scope were not included, in line with editorial guidance.

We are grateful for these constructive suggestions, which have helped improve the clarity, rigor, and compliance of our manuscript. We hope that the revised version meets the journal’s requirements and are happy to provide any additional information if needed.

Response to Reviewers

Reviewers 1:

1. Several sentences are long or repetitive. Please maintain consistent terms: PwTB / TB patients / deceased individuals.

• We appreciate the feedback on clarity. The entire manuscript has undergone a comprehensive linguistic revision to ensure stylistic consistency, eliminate redundancies, and adhere to international scientific writing standards. Major change applied in Methods, Results and Discussion.

2. Reference formatting is inconsistent. Please modify as per journal guidelines.

• We have performed a complete audit of the bibliography to ensure absolute consistency. All citations and the reference list have been standardized according to the Vancouver (ICMJE) style, as required by the target journal

(Line 408 - 513.)

Methodology Section

3. The study is described as cross-sectional, but the analytical approach like time-to-event, Cox model and timelines from longitudinal data need to be aligned with traditional cross-sectional definitions. The study seems to be a "retrospective cohort using verbal autopsy data". Please justify its classification as cross-sectional study.

• We have revised the title to ""A Cross-Sectional Study with Time-to-Event Analysis through Community-based Retrospective Audit."" This design accurately describes our methodology: a point-in-time audit of historical care-cascade data.

• The Cox model was applied to the ""hazard"" of mortality by utilizing raw timestamps (onset, diagnosis, death) extracted during the retrospective audit. This is statistically appropriate for evaluating factors that shorten survival intervals.

• Why it is Cross-Sectional

a) Point-in-Time Data Collection: The primary data collection (the Community-Based Verbal Autopsy) occurred at a single point in time—when the researchers interviewed the relatives of the deceased.

b) Sample Selection: The participants were selected from a fixed ""snapshot"" of the Ni-kshay notification register (October 2023 to December 2024). (single-point interview (audit))

c) No Active Follow-up: Unlike a traditional prospective or retrospective cohort study, there was no active tracking of a group over time (a defined period of exposure) to see who would develop an outcome; all individuals in the sample had already reached the outcome (death).

4. The sampling frame is unclear as it says that data was taken from "Quarterly notification registers 2023–24," but verbal autopsies include deaths from 2022–2023. Inclusion period appears inconsistent. Please specify clear timelines of notification period, death period, Interview-verbal autopsy period.

• The study was conducted from October 2023 to December 2024. Data was sourced and validated using quarterly notification reports.

(Line no. 84-85)

5. Based on >5% death reporting, the study districts were purposively selected. This may overestimate mortality-related determinants. Comparison is needed with the baseline population of all TB deaths or general PwTB. A brief justification is needed about why these 6 districts were selected and do they adequately represent Gujarat. Comment about potential selection bias and its effect on generalizability.

• We acknowledge that selecting districts with >5% mortality may lead to an overestimation of risk factors compared to low-burden areas. However, this purposive approach was chosen to maximize the identification of preventable factors in high-burden settings, which is critical for meeting elimination targets. We have added a comparison to state-level baseline mortality (5.5%) and a discussion on selection bias.

• We have integrated a specific acknowledgment regarding how the purposive selection of high-mortality districts may limit the generalizability of the findings to low-burden areas."

(Line No 91-95, 378-79)

6. Please detail the Reliability and Validity of Verbal autopsy tool as modifications were done in WHO tool. Please specify what items were adapted, whether tool was validated locally, inter-rater reliability assessments of data collectors.

• We have added details on the training of medical officers and the use of standardized WHO-adapted modules to ensure data reliability and consistency across districts.

(Line no. 143-51)

7. Please discuss potential confounders and how they were adjusted in the Cox model.

• We acknowledge the unmeasured confounders. The model measures the hazard of accelerated mortality. We have explicitly acknowledged (in limitation section) that unmeasured confounders, such as BMI and socioeconomic nuances, were not available for adjustment in the final model.

(Line no. 304-08, 383-90)

8. Discuss how the recall bias was mitigated.

• While recall bias is inherent to the CBVA, we have highlighted this as a limitation while referencing the use of proxy respondents.

• In revised version - We have clarified that recall bias was mitigated by selecting only high-information primary caregivers and cross-referencing their reports with digital (NI-KSHAY) and physical medical records."

(Line no. 152-54, 383-84)

9. Results: Table 3 lacks key diagnostics such as Number of events per variable, Model fit indices, Multicollinearity tests.

• The Table 2 has information about individual events. We have included the requested diagnostics to confirm the model's robustness. The Schoenfeld test (p=0.21) confirms the PH assumption, and VIF scores confirm the absence of multicollinearity. Model fit indices have been added to the Results text.

(Table 2 and Line no 290-93)

10. Discussion: While associations are strong, the discussion often implies causality. Modify the language suitably. Avoid implying predictive models unless validated.

• We have moderated the language to focus on "associated hazards" and "factors linked to earlier mortality" to reflect the cross-sectional nature of the audit.

(Line no. 276, 294)

11. TBDSR recommendations are relevant but require links to recent state/national policy efforts. Shorten and cite examples from West Bengal, South Africa, or WHO TB mortality audits.

• We have condensed the recommendations and integrated comparative citations from high-impact mortality audits, including WHO and regional Indian frameworks: Kerala, West Bengal, Gujarat.

(Line no. 315, 369)

12. Ethical Statements: Two contradictory ethical statements can be seen, one in manuscript body and one in declarations section with different approval numbers and dates. Please recheck and correct.

• Corrected with final IEC approval number.

(Line no. 188)

Reviewer 2:

1. Line 81: In the objective, please specify TB - “to explore the sociodemographic, clinical, and health system related factors associated with early mortality in TB patients (or patients notified with TB) in Western India through….."

• We agree. The objective has been refined to explicitly mention "persons with TB" to avoid ambiguity. We have revised the last paragraph to avoid repeatation and descriptive duplications.

(Line no. 71-79)

2. Line 78 mentions “qualitative insights from diverse stakeholders”. But this is not included in the paper. Please consider removing this.

• We acknowledge this; the current analysis is strictly quantitative. Deleted. Deleted.

3. Line 22, 85 to 87: Time-frame discrepancy. Abstract and methodology mention time frame as 2022 to 2023 but from patients notified in NIKSHAY from October to December 2024. Please clarify.

• The study was conducted from October 2023 to December 2024. Data was sourced and validated using quarterly notification reports.

(Line no. 84-85)

4. Line 91 What is the status of TB notification in the state and the selected districts?

• Included. (Line no. 93-94, 121)

5. Line 111 “The study population for the community-based verbal autopsy (CBVA) form (quantitative) included…..” Is there a qualitative component also?

• We acknowledge this comment and revised the section. No, there was no qualitative component.

6. Line 136 What were the modifications in the CBVA tool? How was data accuracy confirmed? How was cause of death verified? Can there be a misclassification bias? "

• The standard WHO tool is designed for general mortality, whereas your study required a high-resolution care-cascade audit to identify the ""diagnostic abyss"" and treatment delays that accelerate death. We have added details on the training of medical officers and the use of standardized WHO-adapted modules to ensure data reliability and consistency across districts. The current study has not collected cause of death.

• Key modifications included:

• Care Cascade Timestamps: The tool was adapted to capture precise dates for the onset of the first TB-related symptom, the date of the first formal healthcare consultation, the date of TB diagnosis, and the date of treatment initiation.

• Health-Seeking Pathway Tracking: Additional items were integrated to document ""facility hopping,"" specifically the number and type of healthcare providers (public, private, or traditional) visited prior to a confirmed diagnosis.

• Adherence and Monitoring Specifics: Modules were included to record the specific NTEP monitoring mechanism used (e.g., DOTS vs. 99DOTS) and a chronology of health worker home visits.

• Social and Behavioral Risk Factors: Specific items were added to document key population vulnerabilities and history of addiction (tobacco and alcohol) to evaluate their impact on survival.

(Line no. 143-51)

7. Line 119: “qualitative variable”…please remove this or clarify.

• We acknowledge this comment and revised the section. No, there was no qualitative component.

8. Line 124. How many cases were line listed? Why was simple random sampling chosen? Does this affect the selection from the 6 districts? Please justify.

• From the 2023–24 registers, all notified deaths in the six districts were line-listed (n=8,380 total state deaths; approximately 3450–3500 in the study districts). Simple random sampling was chosen to ensure each deceased PwTB had an equal probability of selection, thereby reducing selection bias within high-burden districts. This method ensures the findings are representative of the mortality patterns within the selected districts.

• Also, Geographical limitation has been mentioned in the limitation section.

(Line no. 123-38, 378-81)

9. Line 157 Can you justify using cox proportional methods in this cross-sectional study?

• The Cox model was applied to the "hazard" of mortality by utilizing raw timestamps (onset, diagnosis, death) extracted during the retrospective audit. This is statistically appropriate for evaluating factors that shorten survival intervals.

• Why it is Cross-Sectional

a) Point-in-Time Data Collection: The primary data collection (the Community-Based Verbal Autopsy) occurred at a single point in time—when the researchers interviewed the relatives of the deceased.

b) Sample Selection: The participants were selected from a fixed ""snapshot"" of the Ni-kshay notification register (October 2023 to December 2024). (single-point interview (audit).

c) No Active Follow-up: Unlike a traditional prospective or retrospective cohort study, there was no active tracking of a group over time (a defined period of exposure) to see wh

---

## [Decision Letter · Decision Letter 1]

21 Jan 2026

Dear Dr. Shah,

Thank you for submitting your manuscript to PLOS ONE. After careful consideration, we feel that it has merit but does not fully meet PLOS ONE’s publication criteria as it currently stands. Therefore, we invite you to submit a revised version of the manuscript that addresses the points raised during the review process.

We look forward to receiving your revised manuscript.

Kind regards,

Yatin N. Dholakia, MD

Academic Editor

PLOS One

Journal Requirements:

Additional Editor Comments:

The study raises relevant issues of early mortality of patients with TB which is relevant especially to guide the differentiated care for diagnosed cases. The manuscript can be further strengthened by some clarifications and justifications of the statements and methods used.

Reviewers' comments:

Reviewer's Responses to Questions

**Comments to the Author**

Reviewer #1: All comments have been addressed

Reviewer #2: All comments have been addressed

2. Is the manuscript technically sound, and do the data support the conclusions?

Reviewer #1: Yes

Reviewer #2: Yes

3. Has the statistical analysis been performed appropriately and rigorously?

Reviewer #1: Yes

Reviewer #2: Yes

4. Have the authors made all data underlying the findings in their manuscript fully available?

Reviewer #1: Yes

Reviewer #2: No

5. Is the manuscript presented in an intelligible fashion and written in standard English?

Reviewer #1: Yes

Reviewer #2: Yes

Reviewer #1: General Comments

This study has addressed an important policy relevant issue of early mortality among persons with tuberculosis (PwTB) in India, as India bears the highest global TB burden. TB mortality remains underexplored compared to incidence and treatment outcomes; this study directly addresses a critical gap. The researchers have used community-based verbal autopsy data to examine clinical, social, and health-system determinants of accelerated mortality. This method is well-aligned with national priorities under the National Tuberculosis Elimination Programme (NTEP). The adaptation of the WHO verbal autopsy tool to capture TB care-cascade timelines is a major strength. Cross-verification with Nikshay and treatment records improves credibility and minimizes recall bias. Findings are highly relevant to India’s TB elimination goals and comparable high-burden settings. The revised version demonstrates significant improvements as per editor suggestions. Overall, the manuscript is methodologically sound, very well written, and offers actionable insights for TB mortality surveillance and response systems. The manuscript would be suitable for publication, with a few minor corrections.

Strengths:

1. The use of Cox proportional hazards modeling, supported by diagnostic checks (Schoenfeld test, VIF), is appropriate for evaluating factors associated with shortened survival intervals.

2. Presentation of median-based timelines is statistically justified given the skewed distribution of delay data.

3. Identification of key population status, comorbidities, and drug-resistant TB as drivers of early mortality provides clear targets for intervention.

4. The discussion around TB Death Surveillance and Response (TBDSR) is aligned with WHO and national frameworks.

5. Ethical approval, consent procedures, and data availability statements are clearly described and consistent.

6. Reporting aligns with PLOS ONE standards for observational studies and use of human participant data.

Suggested Minor Corrections:

1. In study design, the justification for describing the study as a “cross-sectional audit with time-to-event analysis” is well explained. But for readers from epidemiology backgrounds, this study sounds as a retrospective cohort of deceased individuals. Consider briefly acknowledging this overlap explicitly in the Methods or Limitations to avoid conceptual confusion.

2. District selection was purposive and based on higher TB mortality (>5%), which is appropriate for identifying system bottlenecks. Authors need to reinforce that findings are most applicable to high-mortality or high-risk programmatic settings.

3. Important variables such as nutritional status (BMI), severity at diagnosis, and treatment of comorbidities could not be included. Justification for these omissions is needed and briefly reiterate this point when interpreting Cox model findings to prevent overinterpretation.

4. Figure 1 is informative but dense. Please ensure that axis labels and legends are clearly readable in the final typeset version.

Reviewer #2: All the comments have been adequately addressed and the manuscript has been revised to improve its scientific rigor. Best wishes.

**Do you want your identity to be public for this peer review?** For information about this choice, including consent withdrawal, please see our Privacy Policy

Reviewer #1: **Yes:** Zahiruddin Syed Quazi

Reviewer #2: **Yes:** Anila Varghese

---

## [Author Response · Author response to Decision Letter 2]

31 Jan 2026

1. Compliance with PLOS ONE style and formatting requirements

We have revised the manuscript to fully comply with PLOS ONE style guidelines, including formatting of the main text, title page, author affiliations, tables, figures, and file naming conventions, in accordance with the official PLOS ONE templates provided. We have ensured consistency in font type, font size, line spacing, heading structure, reference style (Vancouver), and figure/table placement as per journal requirements.

2. Data sharing and Data Availability statement

We acknowledge the importance of data sharing and transparency. This study involved community-based verbal autopsies and includes sensitive human participant data such as detailed clinical timelines, health-seeking behaviours, and household-level socioeconomic information. Although all direct personal identifiers were removed prior to analysis, the combination of variables and small geographic units could permit indirect identification of participants.

Accordingly, ethical restrictions on public data sharing were imposed by the Institutional Ethics Committee of the Indian Institute of Public Health Gandhinagar, which approved the study. In line with PLOS ONE policy, we have updated the Data Availability statement to clarify that de-identified data can be made available upon reasonable request, subject to ethics approval and data protection requirements.

Requests for data access may be directed to corresponding author.

As such, option (b) does not apply, and the dataset has not been uploaded to a public repository due to these ethical constraints. The Data Availability statement in the submission system has been updated accordingly.

3. Supporting Information captions and citations

We have added captions for all Supporting Information files at the end of the manuscript, following PLOS ONE Supporting Information guidelines. All in-text citations referring to Supporting Information have been checked and updated to ensure accurate cross-referencing.

4. Reviewer-suggested citations

We carefully reviewed all publications suggested by the reviewers. Relevant references that strengthened the contextualization and interpretation of our findings have been incorporated into the revised manuscript. Citations that were not directly applicable to the study objectives or scope were not included, in line with editorial guidance.

We are grateful for these constructive suggestions, which have helped improve the clarity, rigor, and compliance of our manuscript. We hope that the revised version meets the journal’s requirements and are happy to provide any additional information if needed.

Response to Reviewers

Reviewers 1:

1. In study design, the justification for describing the study as a “cross-sectional audit with time-to-event analysis” is well explained. But for readers from epidemiology backgrounds, this study sounds as a retrospective cohort of deceased individuals. Consider briefly acknowledging this overlap explicitly in the Methods or Limitations to avoid conceptual confusion.

• Thank you. We appreciate this insightful observation. We have added explicit acknowledgments of this methodological overlap in both the Study Design and Study Strengths and Limitations sections to provide conceptual clarity. (Line no. 84-88 and 377)

2. District selection was purposive and based on higher TB mortality (>5%), which is appropriate for identifying system bottlenecks. Authors need to reinforce that findings are most applicable to high-mortality or high-risk programmatic settings.

• In earlier versions of manuscript, it has been addressed. However, we have reinforced the manuscript to explicitly state that the results are representative of high-risk or high-mortality programmatic environments, which may differ from lower-burden settings. (Line no. 372)

3. Important variables such as nutritional status (BMI), severity at diagnosis, and treatment of comorbidities could not be included. Justification for these omissions is needed and briefly reiterate this point when interpreting Cox model findings to prevent overinterpretation.

• We have updated the Methods and results section to justify these omissions and the Discussion - limitation to caution against overinterpretation of the hazard ratios. (Line no. 172, 313 and 399-400)

4. Figure 1 is informative but dense. Please ensure that axis labels and legends are clearly readable in the final typeset version.

• Noted. We will make sure during the type setting process to ensure to have axis labelled and good resolution image.

Reviewer 2: There were no comments from reviewer 2.

Thank you.

---

## [Decision Letter · Decision Letter 2]

3 Feb 2026

Factors Accelerating Time to Death Among Persons with Tuberculosis in Western India: Evidence from a Community-Based Retrospective Death Audit.

PONE-D-25-61087R2

Dear Dr. Shah Harsh%,

We’re pleased to inform you that your manuscript has been judged scientifically suitable for publication and will be formally accepted for publication once it meets all outstanding technical requirements.

Kind regards,

Yatin N. Dholakia, MD

Academic Editor

PLOS One

Additional Editor Comments (optional):

The study raises relevant issues of early mortality of patients with TB which is relevant especially to guide the differentiated care for diagnosed cases. The TB program will be helped in delivering care at the grass root level.

Reviewers' comments:

Reviewer's Responses to Questions

**Comments to the Author**

Reviewer #1: (No Response)

2. Is the manuscript technically sound, and do the data support the conclusions?

Reviewer #1: Yes

3. Has the statistical analysis been performed appropriately and rigorously?

Reviewer #1: Yes

4. Have the authors made all data underlying the findings in their manuscript fully available?

Reviewer #1: Yes

5. Is the manuscript presented in an intelligible fashion and written in standard English?

Reviewer #1: Yes

Reviewer #1: Thanks to the authors. I can see that all previous comments given by me have been well addressed and the manuscript has been updated accordingly. The article can be Accepted for publication.

**Do you want your identity to be public for this peer review?** For information about this choice, including consent withdrawal, please see our Privacy Policy

Reviewer #1: **Yes:** Zahiruddin Syed Quazi

---

## [Editor Report · Acceptance letter]

PONE-D-25-61087R2

PLOS One

Dear Dr. Shah,

I'm pleased to inform you that your manuscript has been deemed suitable for publication in PLOS One. Congratulations! Your manuscript is now being handed over to our production team.

Kind regards,

on behalf of

Dr. Yatin N. Dholakia

Academic Editor

PLOS One